# HYBRID MAMBA–TRANSFORMER DECODER FOR ERROR-CORRECTING CODES

## ABSTRACT

We introduce a novel deep learning method for decoding error correction codes based on the Mamba architecture, enhanced with Transformer layers. Our approach proposes a hybrid decoder that leverages Mamba's efficient sequential modeling while maintaining the global context capabilities of Transformers. To further improve performance, we design a novel layer-wise masking strategy applied to each Mamba layer, allowing selective attention to relevant code features at different depths. Additionally, we introduce a progressive layer-wise loss, supervising the network at intermediate stages and promoting robust feature extraction throughout the decoding process. Comprehensive experiments across a range of linear codes demonstrate that our method outperforms or matches Transformer-only decoders while improving complexity.

## 1 INTRODUCTION

Deep learning–based decoders have achieved remarkable success in decoding error-correcting codes in recent years. Notable examples include Neural Belief Propagation (Nachmani et al., 2018), which learns weights of the message-passing algorithm; Neural Min-Sum (Lugosch & Gross, 2017; Dai et al., 2021a), which approximates the classical min-sum decoder with trainable parameters; Neural RNN decoder (Kim et al., 2018) for convolutional and turbo error correcting codes. Recently, diffusion-based decoders (Choukroun & Wolf, 2022a), which model channel noise as a diffusion process that can be reversed; and Transformer-based decoders (Choukroun & Wolf, 2022b; 2024b; Park et al., 2024; Zheng et al.), which exploit self-attention to capture the code structure, reached state-of-the-art performance in neural decoding. However, despite their individual strengths, these methods either incur a high computational cost, compared to classical decoders, or fail to achieve state-of-the-art performance on some codes.

In this work, we propose a novel hybrid decoder that combines the Mamba architecture (Gu & Dao, 2023) - known for its highly efficient sequential modeling and low runtime latency - with Transformer layers (Vaswani et al., 2017) that provide global receptive fields throughout the codeword. Concretely, we introduce a layer-wise masking strategy within each Mamba block, enabling the model to selectively focus on the most informative subsets of bits at varying depths. To further bolster the training dynamics, we propose a layer-wise loss that provides intermediate supervision at each decoding stage. This auxiliary loss not only promotes better gradient propagation through deep networks but also encourages the extraction of the decoded codeword at each stage enabling intermediate estimation of the decoded codeword.

Extensive experiments on a diverse suite of binary linear block codes, including BCH, Polar, and LDPC codes, demonstrate that our Mamba–Transformer decoder consistently surpasses both Transformer-only decoders and conventional Mamba implementations. We report relative improvements of up to $18\%$ in BER for BCH and Polar codes, and is on par with LDPC codes, while improving inference speed compared to previous works.

## 2 RELATED WORKS

### 2.1 NEURAL DECODERS

In recent years, the study of deep learning-based decoders for error correction codes has emerged as a vibrant and rapidly evolving research area (Gruber et al., 2017). Two broad paradigms have been pursued: model-based architectures, which embed the structure of classical decoding algorithms into neural networks, and model-free architectures, which treat decoding as a purely data-driven mapping.

**Model-based neural decoders** In model-based approaches, the computational graph of a traditional message-passing decoder is reinterpreted as a deep network with trainable parameters. Neural Belief Propagation (NBP) first demonstrated this idea by assigning learnable weights to the edges and messages of the belief propagation algorithm, resulting in a decoder that jointly optimizes its update rules through gradient-based training (Nachmani et al., 2016). Building on NBP, the Neural Min-Sum decoder approximates the classical Min-Sum algorithm by introducing scalar and vector weight parameters into its summation and normalization steps (Lugosch & Gross, 2017; Dai et al., 2021b; Kwak et al., 2022; 2023). This parameterization retains the low-complexity structure of Min-Sum while achieving performance on par with more expensive decoders. To further reduce inference cost, pruning techniques have been applied to compress these networks, systematically removing redundant connections and yielding lightweight variants without significant performance degradation (Buchberger et al., 2020).

**Model-free neural decoders** In contrast, model-free decoders rely solely on the representational power of generic neural architectures. Early work employed fully-connected networks to directly map noisy codewords to their nearest valid codewords, demonstrating feasibility on short block codes (Cammerer et al., 2017). Subsequent studies showed that such networks can scale to moderate block lengths without overfitting (Bennatan et al., 2018). More advanced generative frameworks have also been introduced: diffusion-based decoders model the channel corruption as a forward stochastic process and learn to reverse it via a sequence of denoising steps, achieving impressive gains under various noise conditions (Choukroun & Wolf, 2022a). Meanwhile, Transformer-based decoders exploit self-attention to capture long-range code constraints; notable examples include the Error Correction Code Transformer with its extensions (Choukroun & Wolf, 2022b; 2024a;b;c) and recent variants employing layer-wise masking and cross-message-passing modules to enhance both expressivity and decoding speed (Park et al., 2023; 2024).

### 2.2 MAMBA ARCHITECTURE

In recent years, State-Space Models (SSMs) have attracted considerable attention as an alternative to purely attention-based architectures for sequence modeling, due to their ability to capture long-range dependencies with favorable computational and memory efficiency (Gu et al., 2021a;b). A landmark contribution in this domain is the Structured State Space Sequence (S4) model, which leverages parameterized linear dynamical systems and the HiPPO framework (Gu et al., 2020) to achieve expressive, convolutional representations of sequential data. Building upon S4, subsequent work proposed the Mamba architecture, wherein the SSM's convolutional kernels are dynamically generated as functions of the input sequence (Gu & Dao, 2023; Dao & Gu, 2024). Empirical evaluations demonstrate that Mamba attains inference speeds up to five times faster than comparable Transformer models while scaling seamlessly to input lengths on the order of millions of elements. Moreover, when Mamba is integrated with Transformer layers in a hybrid configuration, the resulting model consistently surpasses both standalone Transformer and S4 architectures in a range of language and time-series benchmarks.

## 3 BACKGROUND

In this section, we formalize the decoding setup for binary linear block codes using the notation of Choukroun & Wolf (2022b). Let $C \subseteq \mathbb{F}_2^n$ be a binary linear block code of length $n$ and dimension $k$, defined by its parity-check matrix $H \in \mathbb{F}_2^{(n-k) \times n}$. A vector $x \in \mathbb{F}_2^n$ is a valid codeword if and only if $Hx = 0$. Transmission occurs over an Additive White Gaussian Noise (AWGN) channel

with Binary Phase-Shift Keying (BPSK) modulation. Under this model, the codeword $x \in \{0,1\}^n$ is mapped to $x_s \in \{\pm 1\}^n \subset \mathbb{R}^n$ and corrupted by Gaussian noise $z \sim \mathcal{N}(0, \sigma^2 I_n)$, yielding the received vector $y = x_s + z$. To enforce invariance to the transmitted codeword and mitigate overfitting, we construct the decoder input from the magnitude of the channel output and its syndrome as in (Bennatan et al., 2018). First, we obtain the hard-decision vector $y_b = \frac{1-\text{sign}(y)}{2} \in \{0,1\}^n$, and then compute the syndrome $s = H y_b \in \mathbb{F}_2^{n-k}$. Finally, we concatenate the amplitude $|y| \in \mathbb{R}^n$ with the syndrome $s$ to form the decoder input $y_{\text{in}} = \begin{bmatrix} |y| & s \end{bmatrix} \in \mathbb{R}^{n+(n-k)}$, which is provided to the proposed Mamba–Transformer decoder.

## 4 METHOD - MAMBA-TRANSFORMER DECODER

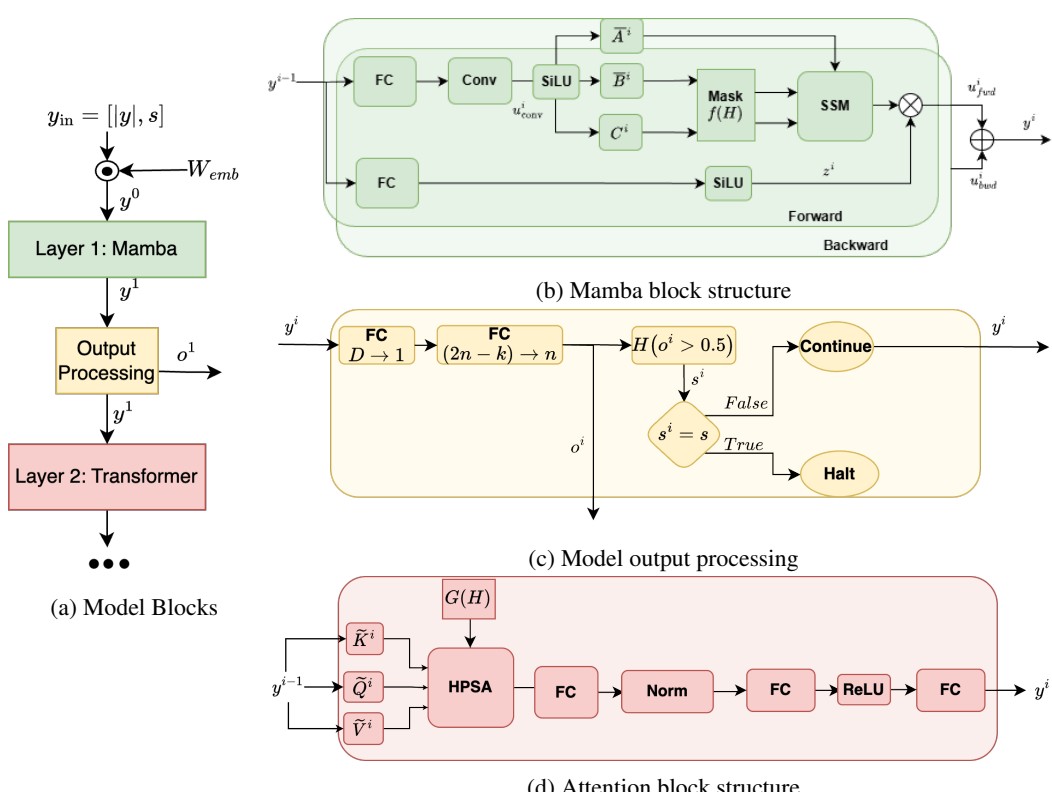

Figure 1: ECCM architecture

The ECCM model is composed of $N_{layers}$ layers, the layers alternate between Mamba layers and attention layers, starting with a Mamba layer. The layers $l_i$ where $i \in 1, 3, ...$ are Mamba layers and the layers $l_i$ where $i \in 2, 4, ...$ are attention blocks. The input to the $l_i$ layer is denoted $y^{i-1}$, with

$$y^0[l,d] = y_{in}[l]W_{emb}[l,d] \tag{1}$$

where $W_{emb}^i \in \mathbb{R}^{L \times D}, l \in [1, L], d \in [1, D]$, where $D$ is the model hidden dimension, $L = 2n - k$, and $\odot$ represents the element-wise multiplication operator. Note that $y^i$ for $i > 0$, will be the output of the $i$-th layer. The architecture uses two masks produced from the parity check matrix, $f(H) \in \mathbb{Z}_2^{(n-k) \times (2n-k)}$ and $g(H) \in \mathbb{Z}_2^{(2n-k) \times (2n-k)}$, which are used by the Mamba layers and attention layers, respectively. The $g(H)$ mask is taken from (Choukroun & Wolf, 2022b):

$$g(H) = \begin{bmatrix} Graph(H) & H^T \\ H & I_{n-k} \end{bmatrix} \tag{2}$$

and $Graph(H) \in \mathbb{R}^{n \times n}$

$$\text{Graph}(H)[i,j] = \begin{cases} 1, & \exists m \in [1, n-k] \text{ such that } H[m,i] = 1 \text{ and } H[m,j] = 1 \\ 0, & \text{otherwise} \end{cases} \tag{3}$$

The $g(H)$ mask ensures that only pairs on the same parity check line are computed, which reduces the complexity of the attention operation and induces knowledge of the code into the model.

The proposed $f(H)$ mask is designed to have the same effect on the SSM process. By applying it to the matrices of the operation, the effect of the input in a specific position only changes the state for bits that are on the same parity check line. Ensuring that interactions only happen along the parity check lines, and across all of the parity check line in contrast to the pairwise interaction of a transformer block.

$$f(H) = [H \ I_{n-k}] \tag{4}$$

### 4.0.1 MAMBA BLOCK

Each Mamba block contains the following operations: First, $y^{i-1}$ is projected with two learnable matrices $W_u^i, W_z^i \in \mathbb{R}^{D \times D}$:

$$u^i = y^{i-1}(W_u^i)^T \tag{5}$$

$$z^i = SiLU(y^{i-1}(W_z^i)^T) \tag{6}$$

where $u^i, z^i \in \mathbb{R}^{L \times D}$ and SiLU is the activation function (Hendrycks & Gimpel, 2023). Then apply 1D-Convolution layer $Conv^i$ to $u^i$ over the sequence length:

$$u_{conv}^i = SiLU(Conv^i(u^i)) \tag{7}$$

where $u_{conv}^i \in \mathbb{R}^{L \times D}$. Then we apply the Selective-State-Space Model Gu & Dao (2023) with modification to the error-correcting code scenario. First, $u_{conv}^i$ is projected to $B^i, C^i \in \mathbb{R}^{L \times S}$ where $S$ is the dimension of the state.

$$B^i = u_{conv}^i W_b^{i T} \tag{8}$$

$$C^i = u_{conv}^i W_c^{i T} \tag{9}$$

Where $W_b^i, W_c^i \in \mathbb{R}^{S \times D}$ are learnable matrices. Then we apply the discretization process (as described in Gu & Dao (2023)) on matrices $B^i$, and $A^i$ where $A^i \in \mathbb{R}^{D \times S}$ which is a learnable matrix. First, generate the $\Delta^i \in \mathbb{R}^{L \times D}$ matrix:

$$\Delta^i = u_{conv}^i (W_\Delta^i)^T \tag{10}$$

where $W_\Delta^i \in \mathbb{R}^{D \times D}$ is a learnable matrix. Second, initialize the tensors $\bar{A}^i, \bar{B}^i \in \mathbb{R}^{L \times D \times S}$,

$$\bar{A}^i[l, d, s] = exp(A^i[d, s]\Delta^i[l, d]) \tag{11}$$

$$\bar{B}^i[l, d, s] = B^i[l, s]\Delta^i[l, d] \tag{12}$$

where $l \in [1, L], s \in [1, S], d \in [1, D]$.

Here, the error-code-specific modification is inserted, using a mask. Generate the mask matrix $f(H)$. Then apply the mask to the matrices $\bar{B}^i$ and $C^i$ creating the matrices $\bar{B}_M^i$ and $C_M^i$, respectively.

$$\bar{B}_M^i[l, d, s] = \begin{cases} f(H)[l, d]\bar{B}^i[l, d, s] & d < (2n - k) \\ 0 & otherwise \end{cases} \tag{13}$$

where $l \in [1, L], s \in [1, S], d \in [1, D]$

$$C_M^i[l, s] = \begin{cases} f(H)[l, s]C^i[l, s] & s < (2n - k) \\ 0 & otherwise \end{cases} \tag{14}$$

Apply the SSM process in which a series of states $h_l \in \mathbb{R}^{D \times S}$ is calculated:

$$h_l[d, s] = \bar{A}[l, d, s]h_{l-1}[d, s] + \bar{B}_M^i[l, d, s]u_{conv}^i[l, d] \tag{15}$$

$$u_{ssm}^i[l, d] = \sum_{i=1}^{S} h_l[d, i]C_M^i[l, i] + R[d]u_{conv}^i[l, d]$$

where $R \in \mathbb{R}^D$ is a learnable vector, and $h_0$ is initialized to a vector of zeros.

Then we apply the gating from Eq.6:

$$u_{fwd}^i = z^i \odot u_{ssm}^i \tag{16}$$

Up to this point, the description was of the processing in the causal direction. Now apply the reverse direction processing in order to achieve a bidirectional Mamba block. Meaning, substitute $y^{i-1}$,$H$ and $u_{fwd}^i$ with $\overleftarrow{y}^{i-1}$, $\overleftarrow{f(H)}$ and $\overleftarrow{u}_{fwd}^i$. And apply Eq.5 through Eq.16 where:

$$\overleftarrow{y}^{i-1}[l,d] = y^{i-1}[L-l,d] \tag{17}$$

$$\overleftarrow{f(H)}[l,d] = f(H)[L-l,d] \tag{18}$$

and $\overleftarrow{u}_{fwd}^i$ is the output of the process. Then calculate $u_{bwd}^i \in \mathbb{R}^{L \times D}$

$$u_{bwd}^i[l,d] = \overleftarrow{u}_{fwd}^i[L-l,d] \tag{19}$$

and then the output of the block $y^i \in \mathbb{R}^{L \times D}$ is calculated.

$$y^i = u_{fwd}^i + u_{bwd}^i \tag{20}$$

## 4.1 TRANSFORMER BLOCK

First, compute the queries, keys, and values $Q^i, K^i, V^i \in \mathbb{R}^{L \times D}$ using the input $y^{i-1}$:

$$Q^i = y^{i-1}W_Q^i + b_Q^i$$
$$K^i = y^{i-1}W_K^i + b_K^i \tag{21}$$
$$V^i = y^{i-1}W_V^i + b_V^i$$

where $W_Q^i, W_K^i, W_V^i \in \mathbb{R}^{D \times D}$ and $b_Q^i, b_K^i, b_V^i \in \mathbb{R}^D$ are learned parameters. These are reshaped into $h$ attention heads with per-head dimension $d_k = D/h$:

$$\tilde{Q}^i, \tilde{K}^i, \tilde{V}^i \in \mathbb{R}^{h \times L \times d_k}. \tag{22}$$

Then apply the HPSA mechanism as described in (Levy et al., 2025):

$$\tilde{O}^i, \alpha^i = \text{HPSA}(\tilde{Q}^i, \tilde{K}^i, \tilde{V}^i, \text{g(H)}), \tag{23}$$

with $\tilde{O}^i \in \mathbb{R}^{h \times L \times d_k}$ and $\alpha^i \in \mathbb{R}^{h \times L \times L}$. The outputs from all heads are concatenated $O^i \in \mathbb{R}^{L \times D}$:

$$O^i = \text{concat}(\tilde{O}^i). \tag{24}$$

Finally, the output of the attention block is computed as $y_a^i \in \mathbb{R}^{L \times D}$:

$$y_a^i = O^iW_O^i + b_O^i \tag{25}$$

where $W_O^i \in \mathbb{R}^{D \times D}$ and $b_O^i \in \mathbb{R}^D$. Then apply layer norm (Ba et al., 2016) to calculate $\tilde{y}_a^i \in \mathbb{R}^{L \times D}$.

$$\tilde{y}_a^i = LayerNorm(y_a^i) \tag{26}$$

Then $y^i \in \mathbb{R}^{L \times D}$ is calculated:

$$y^i = ReLU(\tilde{y}_a^iW_1^i + b_1)W_2 + b_2 \tag{27}$$

where $W_1 \in \mathbb{R}^{D \times 4D}$, $b_1 \in \mathbb{R}^{4D}$, $W_2 \in \mathbb{R}^{4D \times D}$, $b_2 \in \mathbb{R}^D$ are learnable parameters.

## 4.2 MODEL OUTPUT

After each layer, $y^i$ is projected down to $o^i \in \mathbb{R}^n$ using $w_r \in \mathbb{R}^D$,$b_r \in \mathbb{R}^L$,$W_s \in \mathbb{R}^{n \times L}$,$b_s \in \mathbb{R}^n$ which are learnable parameters:

$$o^i = \sigma(W_s(y^iw_r + b_r) + b_s) \tag{28}$$

where $\sigma$ is the sigmoid function. The syndrome is calculated:

$$s^i = H(o^i > 0.5) \tag{29}$$

if $s^i = s$ the processing is stopped - and set $i_{last} = i$, if $s^i \neq s \, \forall \, i \, \in \, [1, N_{layers}]$ set $i_{last} = N_{layers}$ [1]

Note that the model output is an estimate for the input's multiplicative noise, therefore in order to calculate the estimated code-word:

$$\hat{c}^i[l] = \frac{1 - sign((1 - 2o^i[l])y[l])}{2} \tag{30}$$

---

[1] For implementation details in the batch case see Appendix: Early Stopping

## 4.3 LOSS FUNCTION

In order to calculate the loss, first calculate in which bit an error occurred as in Bennatan et al. (2018)

$$z[k] = \frac{1 - sign((1 - 2c[k])y[k])}{2} \tag{31}$$

where $k \in [1, n]$

Then calculate the Binary Cross Entropy (BCE) between $o^i$ and $z$, and sum over all the outputs.

$$Loss = \sum_{i}^{i_{last}} BCE(o^i, z) \tag{32}$$

## 5 EXPERIMENTS

To evaluate the proposed decoder, we train it on four classes of linear block codes: Bose–Chaudhuri–Hocquenghem (BCH) codes (Bose & Ray-Chaudhuri, 1960), Low-Density Parity-Check (LDPC) codes (Gallager, 2003), Polar codes (Arikan, 2009), and MacKay codes. The corresponding parity-check matrices are obtained from Helmling & Scholl (2016). Training samples are generated at six signal-to-noise ratio (SNR) levels, SNR $\in \{2, \dots, 7\}$ dB, and are then added to the generated message to simulate an AWGN channel. We use the zero-codeword in the training process in order to verify that the model doesn't overfit the codewords it sees, by simply changing to random codewords on model evaluation. The Adam (Kingma & Ba, 2017) optimizer was configured with a learning rate of $2.5 \times 10^{-4}$ and decays to $10^{-10}$ following a cosine (Loshchilov & Hutter, 2017) schedule. The training was done with a batch size of 128 and 1000 batches per epoch. In all the experiments we set $D = 128, N_{blocks} = 8, h = 8, S = 128$, where $D$ is the embedding size, $S$ is the Mamba block's state size, $h$ is the number of attention heads, and $N_{blocks}$ is the number of blocks, meaning there are 4 Mamba blocks and 4 attention blocks, the resulting model has a similar number of parameters to previous methods at approx $1.2M$. For evaluation, we simulate test examples at SNR levels of 4dB, 5dB, and 6dB, and report the negative natural logarithm of the bit error rate, $-\ln(\text{BER})$. Each evaluation run is continued until a fixed number of decoding errors, 500, has been observed similar to (Park et al., 2024).

## 6 RESULTS AND DISCUSSIONS

In Tab.1, the results are presented compared to previous methods, For each code 6 methods are presented: BP, ARBP, ECCT, AECCT, CrossMPT, and our own method ECCM. For each, the table shows the negative natural log of the BER at SNR levels 4dB, 5dB, and 6dB. The best method is marked in **bold**, in places where reported results were not available the "-" mark was used. The table shows that ECCM consistently outperforms all the other methods across all BCH codes, and SNR levels. Notably outperforming CrossMPT - with a significant improvement in the decoding of BCH(63,45) code, achieving over $18\%$ in terms of negative natural logarithm of BER, $-ln(\text{BER})$, ECCM shows comparable performance to CrossMPT in the decoding of the Polar(64,48) code, and shows notable improvements in longer Polar codes - achieving up to $7.2\%$ gain in the Polar(128,86) code. While CrossMPT achieves better results in some of the LDPC codes the improvements are modest typically around $4\%$, ECCM achieves better performance in decoding LDPC(49,24) and - comparable to increased - performance on LDPC(121,80). It also outperforms all other models in the MacKay Code, slightly outperforming CrossMPT, which indicates the model is capable of learning very sparse parity-check matrices. Fig 3 shows the performance in terms of BER as a function of SNR for the above methods. It is important to note that integrating ECCM and CrossMPT is possible - by replacing the AECCT transformer blocks, which in theory may close the gap in LDPC codes decoding.

Table 1: Comparison of decoding performance at three SNR values (4, 5, 6) for BP, ARBP (Nachmani & Wolf, 2021), ECCT (Choukroun & Wolf, 2022b), AECCT (Levy et al., 2025), CrossMPT (Park et al., 2024), and ECCM. The results are measured by the negative natural logarithm of BER ($-\ln(\text{BER})$). The best results are highlighted in **bold**. Higher is better.

| Codes | $(N,K)$ | BP 4 | BP 5 | BP 6 | ARBP 4 | ARBP 5 | ARBP 6 | ECCT 1.2M 4 | ECCT 1.2M 5 | ECCT 1.2M 6 | AECCT 1.2M 4 | AECCT 1.2M 5 | AECCT 1.2M 6 | CrossMPT 1.2M 4 | CrossMPT 1.2M 5 | CrossMPT 1.2M 6 | ECCM 1.2M (ours) 4 | ECCM 1.2M (ours) 5 | ECCM 1.2M (ours) 6 |
|---|---|---|---|---|---|---|---|---|---|---|---|---|---|---|---|---|---|---|---|
| BCH | (31,16) | 4.63 / – | 5.88 / – | 7.60 / – | 5.48 | 7.37 | 9.60 | 6.39 | 8.29 | 10.66 | 7.01 | 9.33 | 12.27 | 6.98 | 9.25 | 12.48 | **7.26** | **9.71** | **12.66** |
| | (63,36) | 3.72 / 4.03 | 4.65 / 5.42 | 5.66 / 7.26 | 4.57 | 6.39 | 8.92 | 4.68 | 6.65 | 9.10 | 5.19 | 6.95 | 9.33 | 5.03 | 6.91 | 9.37 | **5.49** | **7.52** | **10.23** |
| | (63,45) | 4.08 / 4.36 | 4.96 / 5.55 | 6.07 / 7.26 | 4.97 | 6.90 | 9.41 | 5.60 | 7.79 | 10.93 | 5.90 | 8.24 | 11.46 | 5.90 | 8.20 | 11.62 | **7.01** | **10.12** | **14.26** |
| | (63,51) | 4.34 / 4.50 | 5.29 / 5.82 | 6.35 / 7.42 | 5.17 | 7.16 | 9.53 | 5.66 | 7.89 | 11.01 | 5.72 | 8.01 | 11.24 | 5.78 | 8.08 | 11.41 | **6.10** | **8.77** | **12.22** |
| Polar | (64,48) | 3.52 / 4.26 | 4.04 / 5.38 | 4.48 / 6.50 | 5.41 | 7.19 | 9.30 | 6.36 | 8.46 | 11.09 | 6.43 | 8.54 | 11.12 | 6.51 | **8.70** | **11.31** | **6.61** | 8.61 | 11.20 |
| | (128,86) | 3.80 / 4.49 | 4.19 / 5.65 | 4.62 / 6.97 | 5.39 | 7.37 | 10.13 | 6.31 | 9.01 | 12.45 | 6.04 | 8.56 | 11.81 | 7.51 | 10.83 | 15.24 | **8.05** | **11.55** | **15.65** |
| | (128,96) | 3.99 / 4.61 | 4.41 / 5.79 | 4.78 / 7.08 | 5.27 | 7.44 | 10.20 | 6.31 | 9.12 | 12.47 | 6.11 | 8.81 | 12.15 | 7.15 | 10.15 | 13.13 | **7.49** | **10.45** | **13.27** |
| LDPC | (49,24) | 5.30 / 6.23 | 7.28 / 8.19 | 9.88 / 11.72 | 6.58 | 9.39 | 12.39 | 5.79 | 8.13 | 11.40 | 6.10 | 8.65 | 12.34 | 6.68 | 9.52 | 13.19 | **6.71** | **9.55** | **13.25** |
| | (121,60) | 4.82 / – | 7.21 / – | 10.87 / – | 5.22 | 8.31 | 13.07 | 5.01 | 7.99 | 12.78 | 5.17 | 8.32 | 13.40 | **5.74** | **9.26** | **14.78** | 5.49 | 8.87 | 14.23 |
| | (121,80) | 6.66 / – | 9.82 / – | 13.98 / – | 7.22 | 11.03 | 15.90 | – | – | – | – | – | – | **7.99** | **12.75** | 18.15 | 7.81 | 12.34 | **18.35** |
| MacKay | (96,48) | – / – | – / – | – / – | 7.43 | 10.65 | 14.65 | – | – | – | – | – | – | 7.97 | 11.77 | 15.52 | **7.98** | **11.84** | **15.70** |

## 7  MODEL ANALYSIS

### 7.1  ABLATION ANALYSIS

Table 2: Ablation analysis: the negative natural logarithm of bit error rate (BER) for our complete method compared with its partial components. Higher values indicate better performance. Highest value is marked in **bold**.

| Experiment | Mamba Mask | Model Layout | Multi-Loss | SNR (dB) 4 | SNR (dB) 5 | SNR (dB) 6 |
|---|---|---|---|---|---|---|
| **Full Method** | $f(H)$ | Transformer & Mamba | True | **7.01** | **10.12** | **14.26** |
| (i) | $g(H)$ | Transformer & Mamba | True | 6.86 | 9.88 | 13.76 |
| (ii) | $f(H)$ | Transformer & Mamba | False | 5.80 | 8.18 | 11.60 |
| (iii) | N/A | Transformer only | True | 6.66 | 9.45 | 13.31 |
| (iv)[2] | N/A | Transformer only | True | 6.64 | 9.19 | 12.69 |
| (v)[3] | $f(H)$ | Mamba only | True | 4.40 | 6.09 | 8.05 |
| (vi)[3] [4] | $f(H)$ | Mamba only | True | 5.22 | 7.17 | 10.07 |

To analyze the contribution of each of the following proposed modifications: combining Mamba and Transformer, using loss from every layer, and the proposed mask for the Mamba layers, variants of the proposed method were trained, removing one modification at a time. The variants were trained on the BCH(63,45) code, with the same hyperparameters as discussed above, excluding experiment (iv) [2]. Note that the number of layers was set to 8 in all experiments, with the exception of experiment (vi), and therefore experiments (iii) and (iv) are on larger models in terms of parameter count (1.6M parameters) relative to ECCM (1.2M parameters) and previous works. In addition, experiment (v) is smaller than the rest at 0.8M parameters, and experiment (vi) was carried out with 12 layers to

---

[2] Experiment (iv) is similar to (iii) but with hyperparameters from ECCT (Choukroun & Wolf, 2022b), $lr = 10^{-4}$, $\eta_{min} = 10^{-6}$

[3] When training a Mamba only model training is unstable, causing gradients to explode. The reported results are the accuracy from the last epoch before the output becomes invalid.

[4] Experiment (vi) is similar to (v) but with 12 layers instead of 8

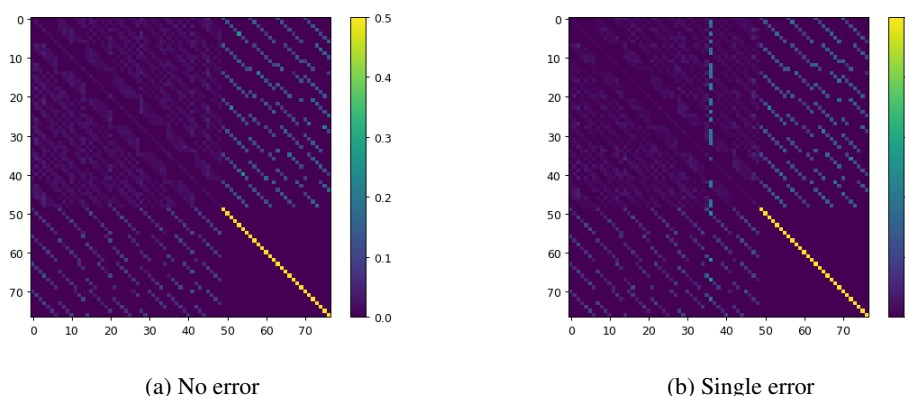

(a) No error                          (b) Single error

Figure 2: Comparison of attention maps sum. (a) without any error (b) with a single error.

complement experiment (v). The same performance evaluation process was used in this study as well. Table 2 shows that each of the proposed modifications contributes to the performance of the final model. The top row shows the full method, and in each subsequent row, one modification is removed to isolate its effect. The "Mamba Mask" indicates which mask was used in the experiment in the Mamba layers, either the proposed mask $f(H)$, or the baseline mask $g(H)$ from ECCT (Choukroun & Wolf, 2022b). In experiment (iii) no Mamba layers are used. The "Model Layout" column indicates whether in the experiment Mamba and Transformer layers, only Transformer layers, or only Mamba layers were used. The "Multi-loss" column is "True" if the loss was computed using the output from each layer, and "False" where it was computed only on the output of the last layer. Experiment (i) shows that using the proposed mask $f(H)$ yields better results than using $g(H)$. Experiment (ii) demonstrates that using the loss from each layer contributes significantly to the proposed model's performance. Experiment (iii) shows that removing the Mamba layers yields worse results, when compared to both experiment (i) and the proposed model, confirming that the modification is an improvement regardless of the mask used.

## 7.2 COMPLEXITY ANALYSIS

The complexity of the proposed method can be separated into the complexity of the transformer block and the complexity of the Mamba block. The complexity of the transformer block is $O((2n - k)D^2 + (2n - k)^2 D\rho(G(H)))$ where $\rho(A)$ is the sparsity of the mask matrix. Moreover, the complexity of the Mamba blocks is $O((2n - k)DS)$, the total complexity of the model is $O(LD(N_{Mamba}S + N_{transformer}(D + L\rho(G(H)))))$, since the $N_{Mamba} = N_{transformer} = \frac{1}{2}N_{blocks}$, we have a significant speedup [5] relative to AECCT and CrossMPT which are $O((2n - k)D^2 + (2n - k)^2(\rho(G(H)))$ and $O((2n - k)D^2 + n(n - k)(\rho((H))))$ (Park et al., 2024) respectively.

## 7.3 ATTENTION SCORE COMPARISON

In order to compare our model's behavior with ECCT (Choukroun & Wolf, 2022b), examination of the internal attention scores of the model's layers in two cases, one where there is no error in the input, and the other where there is a single error in the input. This method reveals how the attention changes in response to error. To visualize the attention across the model, compute the full forward pass of the model with the two inputs, and sum the attention scores across the transformer blocks of the model. For this experiment, evaluate all the layers regardless of whether syndrome condition is met in Eq. 29. Examining the attention maps Fig 2, we can identify four distinct regions corresponding to the structure of the $g(H)$ mask: *magnitude → magnitude* (top-left), *magnitude → syndrome* (top-right),

---

[5]The speedup discussed in this section is regarding only the theoretical complexity. For empirical evidence, and the rationale for its exclusion from the main body of the paper see Appendix: Empirical Processing Time Measurements

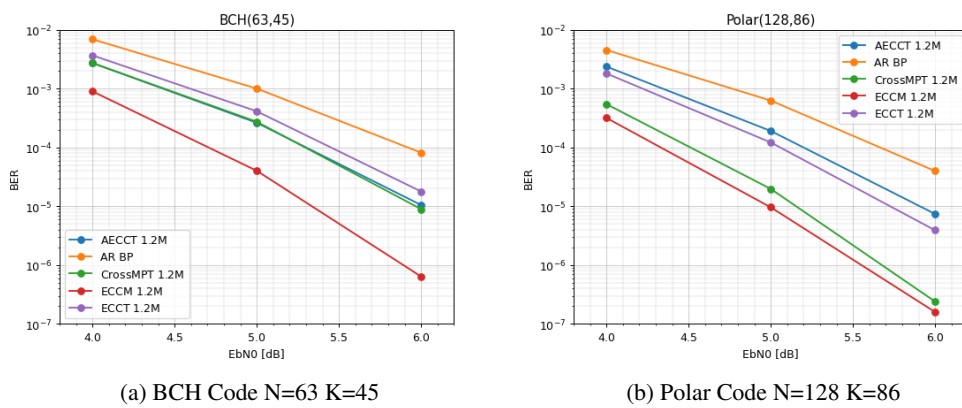

(a) BCH Code N=63 K=45          (b) Polar Code N=128 K=86

Figure 3: BER-SNR performance of ECCM versus baselines, on BCH and POLAR codes

*syndrome → magnitude* (bottom-left), and *syndrome → syndrome* (bottom-right). Each of these regions exhibits different behaviors. Notably, the *syndrome → syndrome* attention is consistently strong, indicating that the model relies heavily on the syndrome. In addition, the *magnitude →  syndrome* attention also remains relatively unchanged regardless of the presence of errors, suggesting that the model treats the syndrome as a reference for interpreting the magnitudes, rather than vice versa. Furthermore, when no error is present, the attention in both the *magnitude → magnitude* and *syndrome → magnitude* regions is low. This implies that the model has learned to infer the presence or absence of errors primarily from the syndrome. However, when an error is present, there is a clear increase in attention across the corresponding column, indicating that the model has learned to examine the entire parity-check line to locate and assess potential errors. In previous analysis on ECCT (Park et al., 2024) the *magnitude→magnitude* and *syndrome→syndrome* relations were less significant leading to the design of the mask in CrossMPT. This analysis shows that ECCM is able to leverage those relations in contrast with previous works.

## 8  LIMITATIONS AND BROADER IMPACTS

**Limitations:** While the proposed ECCM decoder demonstrates strong empirical performance and competitive inference efficiency, several limitations should be noted. First, the model architecture, although designed to generalize across code families, was primarily tested on standard benchmarks with moderate block lengths. Its generalization to very long block codes or non-binary codes remains unverified and may require architectural scaling or retraining. Second, while the hybrid Mamba–Transformer structure improves efficiency over attention-only models, the total model complexity remains non-trivial, and resource-constrained environments (e.g., edge devices) may still face deployment challenges. **Broader Impacts:** Error correction codes are foundational to reliable communication and data storage. The proposed ECCM method improves both the speed and accuracy of decoding. Accuracy improvements can benefit a wide range of technologies, with deep-space transmissions being a notable example, while speed gains may enable learned decoders in real-time systems. However, the black-box nature of learned decoders like ECCM may pose challenges in safety-critical applications where certifiability and interpretability are essential.

## 9  CONCLUSIONS

We introduced ECCM, a hybrid Mamba–Transformer decoder for linear error correction codes. By combining Mamba's efficient sequential modeling with the global context modeling of Transformers, and incorporating parity-check-aware masking and progressive supervision, ECCM achieves state-of-the-art accuracy while maintaining low and improving inference speed. Experimental results across multiple code families demonstrate consistent improvements over existing neural decoders. These findings highlight the potential of hybrid architectures for real-time, high-accuracy decoding, and open the door to further exploration of structured neural models in communication systems.

## 10 USE OF LARGE LANGUAGE MODELS (LLMs)

Large language models (LLMs) were used in this work as an editorial tool, limited to fixing grammar, correcting spelling errors, and improving phrasing. They were not used for research design, data analysis, or drawing scientific conclusions.

## 11 REPRODUCIBILITY STATEMENT

For ease of reproducibility, the code and instructions are provided as supplementary material.

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

# A APPENDIX: COMPARISON TO CLASSICAL DECODERS

For completeness, we are adding comparison to classical decoders. The comparison demonstrates the state of neural decoders in comparison to existing classical methods. We think it shows that neural decoders are on par with classical methods, and their potential advantages - mainly being differentiable outweighs the accuracy degradation.

## A.1 BCH

Table 3: Decoding performance for BCH codes at three SNR values (4, 5, 6). Results are measured by $-\ln(\mathrm{BER})$. Best results in **bold**.

| Codes | $(N, K)$ | BM | | | ECCM (ours) | | |
|---|---|---|---|---|---|---|---|
| | | 4 | 5 | 6 | 4 | 5 | 6 |
| | (63,45) | 4.84 | 6.42 | 8.77 | **7.01** | **10.12** | **14.26** |
| | (31,16) | 4.06 | 5.59 | 7.12 | **7.26** | **9.71** | **12.66** |
| | (63,36) | 4.87 | 7.08 | 9.65 | **5.49** | **7.52** | **10.23** |

On BCH codes, ECCM consistently outperforms the BM baselines across all block lengths. The gains are particularly notable on BCH(63,45), where ECCM achieves more than a 5 dB improvement in $-\ln(\mathrm{BER})$ at high SNR. These results highlight the advantage of combining Mamba and Transformer components with parity-check-aware masking. We do not compare against Ordered Statistics Decoding (OSD), since its significantly higher computational complexity makes it non-comparable in practice.

## A.2 POLAR

Table 4: Decoding performance for Polar codes at three SNR values (4, 5, 6). Results are measured by $-\ln(\mathrm{BER})$. Best results in **bold**.

| Codes | $(N, K)$ | SCL ($L = 32$) | | | CrossMPT | | | ECCM (ours) | | |
|---|---|---|---|---|---|---|---|---|---|---|
| | | 4 | 5 | 6 | 4 | 5 | 6 | 4 | 5 | 6 |
| | (64,48) | 6.56 | **8.85** | 11.28 | 6.51 | 8.70 | **11.31** | **6.61** | 8.61 | 11.20 |
| | (64,32) | **8.13** | **10.68** | **14.07** | 7.51 | 9.97 | 13.31 | 7.84 | 10.30 | 13.40 |

For Polar codes, ECCM narrows the performance gap to strong baselines. On Polar(64,48), ECCM, CrossMPT, and SCL($L = 32$) exhibit very similar results, with differences likely due to simulation randomness rather than fundamental performance. On Polar(64,32),however, ECCM clearly improves over CrossMPT: while CrossMPT lags behind SCL by about 0.6–0.7 nats across SNR levels, ECCM reduces this gap to roughly 0.3–0.4 nats, showing that the proposed method meaningfully closes the distance to the strong SCL baseline.

## A.3 POLAR WITH CRC

Table 5: Decoding performance for Polar(64,32) with 16 bit CRC at three SNR values (4, 5, 6). Results are measured by $-\ln(\mathrm{BER})$. Best results in **bold**.

| Method | 4 | 5 | 6 |
|---|---|---|---|
| ECCM | **7.86** | **10.31** | **13.74** |
| CA-SCL(L=32) | 6.20 | 9.27 | 13.73 |

CRC-Aided Successive Cancellation List (CA-SCL) is considered the SOTA for polar codes. However, it requires additional redundancy bits (the CRC itself), which effectively changes the underlying error

correcting code. Therefore, we treat this experiment as a separate benchmark. In order to compare our method with CA-SCL, we trained a model on a Polar(64,32) where the message included a 16 bit CCITT CRC. The results, including the reference performance of a CA-SCL decoder, are presented in Table 5. From the results, it is apparent that ECCM achieves better accuracy than CA-SCL, which indicates that ECCM is better for CRC aided decoding, although further research is needed to reach a definitive conclusion.

## A.4   LDPC

Table 6: Decoding performance for LDPC(121,80) code at three SNR values (4, 5, 6). Results are measured by $-\ln(\mathrm{BER})$. Best results in **bold**.

| Codes | $(N, K)$ | Layered BP (L=50) | | | Layered BP (L=5) | | | CrossMPT | | | ECCM (ours) | | |
|---|---|---|---|---|---|---|---|---|---|---|---|---|---|
| | | 4 | 5 | 6 | 4 | 5 | 6 | 4 | 5 | 6 | 4 | 5 | 6 |
| LDPC | (121,80) | 7.19 | 11.01 | 16.74 | 6.00 | 8.96 | 13.43 | **7.99** | **12.75** | 18.15 | 7.81 | 12.34 | **18.35** |

For LDPC(121,80), ECCM delivers highly competitive performance compared to both Layered BP and CrossMPT. As expected, increasing the number of BP iterations (L=50 vs. L=5) improves performance, but ECCM and CrossMPT outperform both BP variants across all SNR values. CrossMPT has a slight edge at lower SNRs, while ECCM surpasses it at high SNR, achieving the best result at 6 dB (18.35 vs. 18.15). This shows that ECCM scales well to structured, high-rate LDPC codes, even when compared to specialized iterative decoding methods.

# B  APPENDIX: EARLY STOPPING

## B.1  BATCH IMPLEMENTATION

---

**Algorithm 1** Batch Early Stopping

---

1: $i_{last_b} \leftarrow -1$

2: $o_b^0 \leftarrow \Phi(y_b^i)$
3: $c_b^0 \leftarrow \Gamma(o_b^i)$
4: **for** each $s_i$ in $N$ **do**
5:    **if** $\text{All}(c)$ **then**
6:       **break**
7:    **end if**
8:    $y_b^i[\neg c] = s_i(y_b^i[\neg c])$
9:    $o_b^i \leftarrow \Phi(y_b^i)$
10:   $c_b^i \leftarrow \Gamma(o_b^i)$
11: **end for**

---

Where:

- $\Phi(h_b)$ is Eq. 28 applied to each member of the batch independently.

- $\Gamma(h_b)$ is Eq. 29 applied to each member of the batch independently.

- $s_i$ is the function that applies the $i$-th layer to each member of the batch independently.

- $o_b^i \in \mathbb{R}^{b \times n}$ is the output at layer $i$ for each member of the batch.

- $y_b^i \in \mathbb{R}^{b \times L \times D}$ is the hidden representation at layer $i$ for each member of the batch.

- $c_b^i \in \{0, 1\}^b$ is the vector indicating which batch elements are corrected at layer $i$.

## B.2  LAYER USAGE

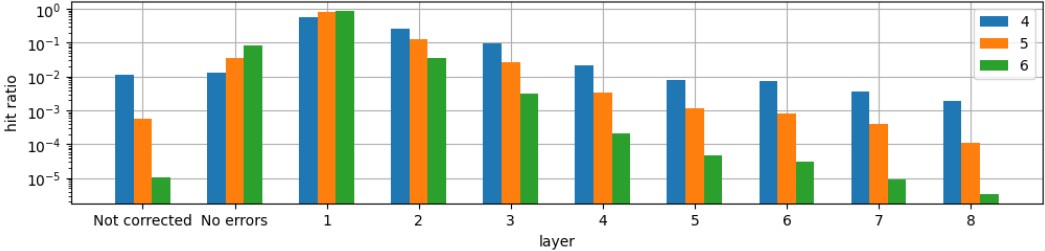

Figure 4: Layer usage statistics for decoding BCH(63,45) messages

In order to evaluate the effectiveness of the early stopping mechanism we ran a BER evaluation experiment as described in Section 5. Instead of tracking BER we track $i_{last}$, Figure 4 shows the rate at which a layer was reached, for BCH(63,45). We can see that most messages are corrected after the first two layers, and that a non negligible but small number of messages start with no error. In addition, we see that later layers exhibit diminishing returns in term of corrected messages and the last layer corrects significantly less messages than the total non corrected messages.

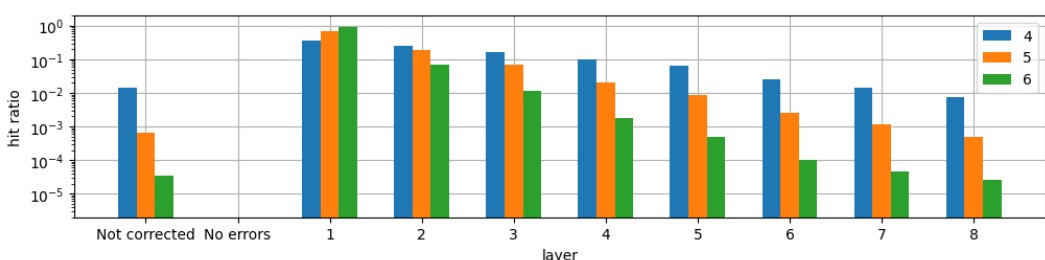

Figure 5: Layer usage statistics for decoding POLAR(128,96) messages

In Figure 5 we can see that unlike the case for BCH(63,45), in POLAR(128,96) all messages required some correction. This is expected, since the probability of an error increases with the length of the message. We can see that most message are corrected and the first layers are much more dominant in the operation of the network, with later layers exhibiting diminishing returns. A

## C    Appendix: Empirical Processing Time Measurements

Validating the claim that the suggested method is more efficient, the following experiments were performed. For each of the methods ECCT, AECCT, CrossMPT, and ECCM - on the same machine apply inference on batches of codewords, with the same number of batches, measure the time that the process took and divide by the number of codewords generated. The process was performed with a batch size of 512 and 200 batches, on a machine with a NVIDIA GeForce RTX 4090 GPU and a Intel i9-14400K. For this test, the Early Stopping feature of the model was disabled.

Table 7: Average inference speed in $\mu s$ for ECCT, AECCT, ECCM, and CrossMPT. Lower is better.

| Code | ECCT | AECCT | CrossMPT | ECCM |
|------|------|-------|----------|------|
| LDPC (121,60) | 332.23 $\mu s$ | 358.81 $\mu s$ | 289.11 $\mu s$ | **260.42** $\mu s$ |
| Polar (128,96) | 315.56 $\mu s$ | 332.23 $\mu s$ | 272.41 $\mu s$ | **221.98** $\mu s$ |

On the same machine, a similar test was performed but instead of testing on the different methods, the test was performed only on ECCM and at different SNRs [dB]: 4, 5, 6. The intention of this test is to show that the early-stopping feature is meaningful for inference runtime. The test results (Table 8 indicates that a partial model can still be used without major performance degradation for SNRs 5[dB] and 6[dB], since the speedup in these cases is coming from layers of the model not being used.

Table 8: Average inference speed at diferent SNRs. Lower is better, the best time is marked in bold.

| Code | No Early Stopping | 4 [dB] | 5 [dB] | 6 [dB] |
|------|-------------------|--------|--------|--------|
| LDPC (121,60) | 260.42 $\mu s$ | 119.78 $\mu s$ | 74.18 $\mu s$ | **53.72** $\mu s$ |
| Polar (128,96) | 221.98 $\mu s$ | 86.37 $\mu s$ | 57.83 $\mu s$ | **46.75** $\mu s$ |

On the same machine, an additional test was performed. Using the same ECCM model trained on Polar(128,96) and early stopping feature turned off, the above experiment ran this time tracking accuracy in addition to processing time. The results are displayed in Table 9, with the exception of the last all layers contribute significantly, though with diminishing returns from the 5-th layer onward.

Table 9: Performance/Layer Count Table showing the Accuracy and Latency for decoding Polar(128,96)

| Layers | 4 [dB] | 5[dB] | 6[dB] | Latency |
|--------|--------|-------|-------|---------|
| 1 | 4.29 | 5.44 | 6.93 | 41 $\mu s$ |
| 2 | 4.88 | 6.39 | 8.86 | 58 $\mu s$ |
| 3 | 5.60 | 7.72 | 10.6 | 95 $\mu s$ |
| 4 | 6.07 | 8.65 | 11.52 | 111 $\mu s$ |
| 5 | 6.51 | 9.39 | 12.55 | 149 $\mu s$ |
| 6 | 7.09 | 10.13 | 12.90 | 167 $\mu s$ |
| 7 | 7.32 | 10.26 | 13.23 | 205 $\mu s$ |
| 8 | 7.65 | 10.51 | 13.28 | 222 $\mu s$ |
| Ref | 7.49 | 10.45 | 13.27 | 86-47 $\mu s$ |

The empirical processing-time measurements reported in Table 7 and 8 are included in the appendix because they do not fully reflect the performance characteristics of a real-world deployment. In practice, a hardware-oriented implementation could exploit architectural properties that are not available in our software-based general purpose GPU evaluation-for example, by specifically leveraging the trinary nature of the HPSA representation. This is particularly relevant for AECCT and ECCM, as this property could be used to achieve substantial acceleration in custom hardware. Nevertheless,

we include these measurements as they represent the most faithful comparison attainable within our current experimental constraints.

