# OpenReview forum: "Hybrid Mamba–Transformer Decoder for Error-Correcting Codes"
_ICLR.cc/2026/Conference — Submitted to ICLR 2026_

### Official Review · Reviewer_RSom · 2025-10-31

**Soundness:** 3
**Presentation:** 4
**Contribution:** 3
**Rating:** 6
**Confidence:** 2

**Summary:**

The paper presents a hybrid Mamba-Transformer architecture for decoding error correcting codes in wireless transmissions. This scheme is ablation tested by removing components - only the Mamba, the transformer are tested against the use of both. The unique characteristics of SSMs allowing retention of very long context complements the transformer's ability to fit to prior training data in a shorter context window with high fidelity, allowing high decode fidelity here. The approach is contrasted against prior art using machine learning methods for error correcting code decoding.

**Strengths:**

The paper's approach outperforms the state of the art and is shown to be the optimal design given the components the authors have used - ablations demonstrate that removal of components from this system degrades performance. The bit-error based binary cross entropy loss ensures decoder training remains stable.

**Weaknesses:**

The given architecture seems much more complex than prior art, and given that channel noise decoding and wireless systems are real-time systems, I would ask what the latency is and whether this high-performance algorithm is suitable for the applications the authors have envisioned.

I also believe the paper would benefit from a hyperparameter sweep of the network, as it is the experiments seem to have been conducted for a given parameter set but we do not know if that is the optimal one. A balance of latency and quality would be necessary for these applications, and a Pareto plot of architecture/parameter count normalized by latency plotted against quality may be illustrative.

**Questions:**

Would the authors say that this method remains unproven for real applications, given the complexity of transformers? Where do they see these approaches being used for real world deployments? Server-side decoder deployments may be unrealistic given latency and transmission requirements for wireless systems, is this intended for the edge?

---

> ### Author Response · Authors · 2025-11-20
>
> We agree that this method remains unproven for widespread commercial applications until specialized hardware is developed. Our work provides the algorithmic foundation and demonstrates the efficiency gain required to motivate future dedicated hardware implementations.
>
> We believe our current work can be enough to justify development of custom hardware and deployment on the receiving device. For use cases such as a cellular base station (gNB), a high-end satellite receiver, or even sophisticated Wi-Fi access points, where power constraints are less strict, our method might be even viable with off the shelf solutions. For low-end edge devices, further work is required specifically in tailoring a low-power solution. However, in order to utilise the full potential of our work, integration with an end to end ML based transceiver where the benefits of having a differentiable decoder can truly shine.
>
> Due to resource limits a very limited hyperparameter sweep was performed, we did not discuss it in the paper since we did not deem it exhaustive enough for any meaningful conclusions.
>
> A Table with the requested size and performance relationship, was added in Appendix: Empirical Processing Time Measurements, Table 9. The table shows the accuracy and latency as a function of the number of layers. While there are other hyperparameters that can control the size of the network this is the only one we could demonstrate a significant range of values without expensive retraining, and in the context of the review discussion.
>
> Adding the table here as well:
> ### Performance / Layer Count for Polar(128,96)
> Accuracy at SNR = 4, 5, 6 dB and corresponding latency.
>
> | **Layers** | **4 dB** | **5 dB** | **6 dB** | **Latency** |
> |-----------|----------|----------|----------|--------------|
> | 1 | 4.29 | 5.44 | 6.93 | 41 μs |
> | 2 | 4.88 | 6.39 | 8.86 | 58 μs |
> | 3 | 5.60 | 7.72 | 10.6 | 95 μs |
> | 4 | 6.07 | 8.65 | 11.52 | 111 μs |
> | 5 | 6.51 | 9.39 | 12.55 | 149 μs |
> | 6 | 7.09 | 10.13 | 12.90 | 167 μs |
> | 7 | 7.32 | 10.26 | 13.23 | 205 μs |
> | 8 | 7.65 | 10.51 | 13.28 | 222 μs |
> | **Ref** | 7.49 | 10.45 | 13.27 | 86–47 μs |

---

### Official Review · Reviewer_j8n8 · 2025-10-31

**Soundness:** 3
**Presentation:** 1
**Contribution:** 3
**Rating:** 4
**Confidence:** 4

**Summary:**

This paper presents ECCM, a hybrid decoder that integrates Mamba and Transformer architectures for error correction code (ECC) decoding, marking the first use of Mamba’s state-space model in this domain. By combining Mamba’s efficient sequential processing with Transformer’s global attention, the method improves computational efficiency and captures long-range bit dependencies. The authors introduce a new masking strategy for Mamba layers, alternate Mamba and Transformer blocks, and apply progressive layer-wise loss, showing improved performance.

**Strengths:**

The paper is novel as it introduces the first use of Mamba architecture for ECC decoding. The approach shows consistent improvements across various code families, with experiments clearly validating the impact of each component including the Mamba architecture, masking strategy, and layer-wise loss, supporting the authors’ design choices.

**Weaknesses:**

Despite the interesting idea, the paper has some clarity issues that make the method difficult to understand and implement correctly. These issues are detailed further in the questions section.

**Questions:**

1. In Sec: 4.2 and 4.3 the authors state that if s_i=s the sample stops, and later outputs are “not calculated” and “not summed” in the loss. Could you add a short pseudocode block showing how this is implemented per sample within a batch during training and testing? And, could you please add a brief layer-usage summary showing where samples stop (i_"last" ) in particular, how many samples terminate before the final block?
2. For Table 2, could you explicitly state the number of blocks used in the ‘Transformer only’ and ‘Mamba only’ settings?
3. The authors train the Hybrid model with Adam at 2.5×10^-4and cosine annealing down to  10^-10(1000 batches/epoch). Could you briefly justify diverging from ECCT/CrossMPT LR/schedules? For Table 2, could you confirm that the ‘Transformer-only’ used the same LR/schedule? For comparability, please add a control run using the ECCT/CrossMPT learning rate and schedule.

---

> ### Author Response · Authors · 2025-11-20
>
> We have added a new section, Appendix: Early Stopping, which includes the requested:
> 1. __A short pseudocode block__ detailing how samples are managed (removed/masked) within a batch when the stopping condition ($s_i = s_{target}$) is met during both training and testing.
> Adding the batch early stopping psuedo code here as well.
> **Batch Early Stopping Pseudo-code**
>
> ```pseudo
> i_last_b ← -1̅
> o_b⁰ ← Φ(y_bᶦ)
> c_b⁰ ← Γ(o_bᶦ)
>
> for each s_i in N:
>     if All(c):
>         break
>
>     y_bᶦ[¬c] = s_i(y_bᶦ[¬c])
>     o_bᶦ ← Φ(y_bᶦ)
>     c_bᶦ ← Γ(o_bᶦ)
> ```
> Where:
>
> - $\Phi(h_b)$ is Eq. (28) applied to each member of the batch independently.
> - $\Gamma(h_b)$ is Eq. (29) applied to each member of the batch independently.
> - $s_i$ is the function that applies the $i$-th layer to each member of the batch independently.
> - $o_b^i \in \mathbb{R}^{b \times n}$ is the output at layer $i$ for each member of the batch.
> - $y_b^i \in \mathbb{R}^{b \times L \times D}$ is the hidden representation at layer $i$ for each member of the batch.
> - $c_b^i \in \{0,1\}^b$ is the vector indicating which batch elements are corrected at layer $i$.
> 2. __A brief layer-usage summary__ to illustrate the distribution of termination layers ($i_{\text{last}}$), providing empirical insight into how many samples terminate before the final block and demonstrating the effectiveness of the mechanism in reducing average computation, the layer-usage summary doesn't fit the comment format therefore it was not added here.
>
> We confirm that for all models tested in Table 2 (Ablation Study), including the 'Transformer only' and 'Mamba only' settings, we consistently used 8 blocks (i.e., $N_{\text{blocks}}=8$). This ensures a fair comparison of the architectural component performance, isolating the effect of the layer type. We have updated the main text discussion for Table 2 to explicitly state this number, note that an additional experiment was added which uses more blocks.
>
> The change in our learning rate (LR) and schedule (Adam at $2.5 \times 10^{-4}$ with cosine annealing down to $10^{-10}$) was the result of a limited hyperparameter sweep optimization. We confirm that all experiments in the ablation study (Table 2) used this new, optimized LR/schedule for consistency. To address the reviewer's concern about comparability, we have added a new control row to Table 2. This row shows the performance of the “Transformer only” model when trained using the original ECCT/CrossMPT LR and schedule, which confirms our hyperparameter choice.
>
> Adding the updated ablation results table:
> ### Ablation Study Results
>
> | Experiment | Mamba Mask | Model Layout | Multi-Loss | SNR 4 dB | SNR 5 dB | SNR 6 dB |
> |-----------|------------|--------------|------------|----------|----------|----------|
> | **Full Method** | `f(H)` | Transformer & Mamba | True  | **7.01** | **10.12** | **14.26** |
> | (i)   | `g(H)` | Transformer & Mamba | True  | 6.86 | 9.88 | 13.76 |
> | (ii)  | `f(H)` | Transformer & Mamba | False | 5.80 | 8.18 | 11.60 |
> | (iii) | N/A | Transformer only | True | 6.66 | 9.45 | 13.31 |
> | (iv)\* | N/A | Transformer only | True | 6.64 | 9.19 | 12.69 |
> | (v)\*\* | `f(H)` | Mamba only | True | 4.40 | 6.09 | 8.05 |
> | (vi)\*\*† | `f(H)` | Mamba only | True | 5.22 | 7.17 | 10.07 |
>
> ---
>
> \* Experiment (iv) is similar to (iii) but uses ECCT hyperparameters: `lr = 1e-4`, `η_min = 1e-6`
> \*\* When training a Mamba-only model, training is unstable and gradients explode. Reported results are from the last valid epoch.
> † Experiment (vi) is similar to (v) but uses 12 layers instead of 8.

---

### Official Review · Reviewer_v6DD · 2025-11-01

**Soundness:** 3
**Presentation:** 1
**Contribution:** 3
**Rating:** 4
**Confidence:** 3

**Summary:**

This paper proposes the Error-Correcting Code Mamba-Transformer (ECCM), a hybrid neural decoder architecture for linear block codes. The design alternates between Mamba (State-Space Model) layers and standard Transformer attention layers, aiming to leverage the linear-time complexity of Mamba for sequential modeling while retaining the global contextual modeling capabilities of the Transformer.

The authors claim three primary contributions: 1) The **hybrid Mamba-Transformer architecture** itself, 2) a new **layer-wise mask for Mamba layers,** **$f(H)$,** derived from parity check matrix, 3) a **progressive layer-wise loss** **function** combined with a syndrome-based early-exit mechanism to stabilize training and reduce inference latency.

Experiments are conducted on BCH, Polar, and LDPC codes. The results show that the proposed ECCM model achieves performance that is on par with, or in some cases (notably BCH) superior to, existing Transformer-only decoders, while offering improved computational complexity.

**Strengths:**

1. **Novel and Timely Architecture:** The integration of Mamba, a recent and efficient SSM, into the ECC decoding problem is a new architecture for model-free channel decoders. The hybrid approach achieves strong decoding performance.
2. **Strong Empirical Performance:** The model achieves highly competitive, and in the case of several BCH codes, state-of-the-art performance compared to strong Transformer-based baselines like AECCT and CrossMPT. This validates the practical efficacy of the paper’s approach.

**Weaknesses:**

1. **Inconsistencies, Typos, and Presentation Issues:** The paper suffers from many typographical errors and presentation issues that reduce clarity and undermine its professional quality.
    - **Formulation and Notation Errors**
        - Eq (30) & (31): The calculation of error vector $z$ and the estimated codeword $\hat{c}^{i}$ should reference the original received vector $y[]$, not the model input $y_{in}[]$ since $y_{in}=[|y|, s]$.
        - Eq (28): The output dimension $N$ of $o^{i}$ should be $n$, given its relationship to the error vector $z$.
        - Eq (4): The notation uses a semicolon ($;$), which conventionally denotes vertical concatenation, whereas the operation required by the stated dimensions is horizontal concatenation.
        - Eq (31): The iterator should use a different variable than $n$ (e.g., $m$), and the upper value should be $n$ (code length).
        - Eq (28) and (29): The paper used the following rule for hard-decision vector: positive y → 0 (binary) and negative y → 1 (on page 3). Because of this rule, it seems that (28) should be $s^i = H \cdot (o^i < 0.5)$ instead of $s^i = H \cdot (o^i > 0.5)$.
    - **Figure-Text Inconsistency**

        In Figure 1(b), the Mamba block diagram is misleading. It correctly shows a SiLU activation for the $z$ path, but omits the SiLU activation for the $u$ path after the Conv block. This contradicts the text, as Eq (7) explicitly states $u_{conv}^{i}=SiLU(Conv^{i}(u^{i}))$.

    - **Poor Presentation**
    In Figure 1(b), the final output label ($y^i$) is poorly rendered and overlaps with the output arrow, demonstrating a lack of attention to detail and reducing the diagram's readability.
Definition discrepancy in Mamba mask $f(H)$: The mathematical formulation in Eq (13) and (14) is misleading. The paper’s Mamba mask $f(H)$ is defined with dimensions $(n-k) \times (2n-k)$, yet the paper applies it using indices $d $ and $s $ (model/state dimensions), as seen in $f(H)[l, d]$ and $f(H)[l, s]$. This indexing is dimensionally inconsistent with the mask's definition.
3. **Lack of empirical runtime metrics:** The conclusion states that ECCM achieves *“low and improving inference speed”*. However, the main text provides only the big-O complexity (Section 7.2) without any empirical latency, throughput, or FLOPs.

**Questions:**

1. **Runtime evidence:** Were empirical measurements (e.g., samples/sec, latency, GPU memory) taken to support the “*low and improving inference speed*” claim? If so, please include them or provide a runtime benchmark appendix.
2. **Clarification on the SCL Baseline for Polar Code Comparison:** In Appendix 12.2 and Table 4, the paper compares ECCM and CrossMPT against a classical "SCL (L=32)" baseline. Based on this, the authors conclude that ECCM "meaningfully closes the distance to the strong SCL baseline". However, the baseline is not specified as being CRC-Aided (CA-SCL), which is the standard high-performance SCL decoder configuration in modern literature. Could the authors clarify whether the "SCL (L=32)" baseline used in Table 4 includes a CRC? If it does not, how can the claim of closing the gap against a "strong SCL baseline" be justified, given that a ‘plain’ SCL is a significantly weaker baseline than the standard CA-SCL?

---

> ### Author Response · Authors · 2025-11-20
>
> In our case we would argue that theoretical complexity is the most relevant factor for evaluating decoder algorithms. Since for any practical use a dedicated hardware implementation will be required, we can only provide flawed measurements on a general-purpose GPU. Although we agree that supporting the claim "low and improving inference speed,” with some empirical data is important. Therefore, we have conducted the requested measurements and included the results in a new section: Appendix: Empirical Processing Time Measurements.
>
> The table comparing the different methods processing time is added here as well:
>
> ### Average inference speed in μs for ECCT, AECCT, CrossMPT, and ECCM
> *Lower is better.*
>
> | **Code**        | **ECCT**      | **AECCT**     | **CrossMPT** | **ECCM**       |
> |-----------------|---------------|---------------|--------------|----------------|
> | LDPC (121,60)   | 332.23 μs     | 358.81 μs     | 289.11 μs    | **260.42 μs**  |
> | Polar (128,96)  | 315.56 μs     | 332.23 μs     | 272.41 μs    | **221.98 μs**  |
>
>
> Clarification regarding the SCL polar benchmark and CRC aided decoding:
> The "SCL (L=32)" baseline used in Table 4 does not include a CRC. We used it as the benchmark, because the addition of a CRC fundamentally changes the code structure, e.g. decreasing the effective bits per message. Our comparison is thus against the strongest classical decoder for this Polar code without CRC, the SCL decoder. In addition, the proposed method in theory at least can leverage additional or accidental structure in the data without need for change - which means in the case of CRC aided decoding training a model specifically for that case is required. Which brings us to the next point:
> New CA-SCL Benchmark: We recognize that CRC-Aided Polar Codes are a high interest use case. Therefore, we have trained and evaluated ECCM on a Polar code with an added CRC. These new results, including a comparison against the CA-SCL decoder, are now included in the Appendix: Comparison to Classical Decoders, Polar with CRC subsection.
> The table is added here is well:
> ### Decoding performance for Polar(64,32) with 16-bit CRC
> Results shown for SNR = 4, 5, 6, measured as \(-\ln(\mathrm{BER})\).
> Best results in **bold**.
>
> | **Method**        | **4**   | **5**   | **6**   |
> |-------------------|---------|---------|---------|
> | ECCM              | **7.86** | **10.31** | **13.74** |
> | CA-SCL (L=32)     | 6.20    | 9.27    | 13.73   |
>
> We thank the reviewer for the detailed feedback on the paper's presentation, which has significantly improved clarity.
>
> All noted typographical errors and inconsistencies (including those for Equations (4), (13), (14), (28), (30), (31), and Figure 1(b) diagram clarity/labels) have been fully corrected in the revised manuscript. We are confident the revised paper is now free of these issues.
> Hard-Decision Rule (Eqs (28) & (29)): We appreciate the close scrutiny. Our rule is: positive $y \to 0$ (binary) and negative $y \to 1$. Our error-output-based decoding strategy, which is consistent with prior work like ECCT, defines a logical one (1) in the output vector $o$ as indicating a detected error that needs correction. In the evaluation expression, $(1 - 2o^i[k])y[k]$, when $o^i[k]=1$, the expression simplifies to $-y[k]$, which correctly flips the sign of the received symbol to correct the detected error. Therefore, the formulation for the estimated codeword $\hat{x}$ remains correct as written, given our definition of the error vector $o$.

---

### Official Review · Reviewer_nwxx · 2025-11-03

**Soundness:** 2
**Presentation:** 2
**Contribution:** 2
**Rating:** 2
**Confidence:** 5

**Summary:**

This paper proposes a hybrid deep learning-based decoder for error-correcting codes, combining the Mamba architecture with Transformer layers. The proposed model aims to leverage Mamba's efficient sequential modeling while maintaining the global contextual modeling of Transformers. The method is evaluated on a range of binary linear block codes, including BCH, Polar, LDPC, and MacKay codes. The authors report improved decoding performance and inference speed relative to previous neural network-based decoders, such as Transformer-only architectures and the CrossMPT model.

**Strengths:**

The combination of Mamba layers with Transformer layers is novel in the context of ECC decoding. This hybrid architecture appears to improve both decoding performance and computational efficiency relative to previous machine learning-based approaches.

**Weaknesses:**

I also reviewed this paper for NeurIPS (Reviewer ID: u6gT). This submission appears essentially identical to the NeurIPS version, which was, in my view, justifiably rejected.

While the paper is technically correct (though it could have been written more clearly) and the proposed hybrid architecture is interesting, its contribution remains limited in both novelty and impact. The literature on machine learning-based decoders for error-correcting codes is now very extensive, and this work does not meaningfully advance the state-of-the-art, either in methodology or insight.

Overall, the paper is solid but rather incremental. It would be more appropriate for a specialized or mid-tier venue rather than a top-tier conference like ICLR, which typically expects stronger conceptual contributions or more significant performance improvements (the proposed decoder does not beat state-of-the-art decoders).

**Questions:**

I do not have any questions. The paper is technically correct, but in my opinion, the contribution is too limited to warrant acceptance.

---

### Meta-Review · Area_Chair_vJEG · 2026-01-16

**Summary:**

[AC: The initial reviews are mostly negative: 2, 4, 4, 6. The authors dispute that the reviewer giving the lowest score 2 may be biased by his prior experience with the authors’ previous submission to NeurIPS. The author with the highest score has low confidence. It also appears that his comments are not fully addressed. Given that some concerns remain not fully cleared and that none of the reviewers responded during the rebuttal period, this paper needs more work.]

Reviewer nwxx (2: reject, not good enough; 5: You are absolutely certain about your assessment.)

-	I also reviewed this paper for NeurIPS (Reviewer ID: u6gT). This submission appears essentially identical to the NeurIPS version, which was, in my view, justifiably rejected.

-	While the paper is technically correct (though it could have been written more clearly) and the proposed hybrid architecture is interesting, its contribution remains limited in both novelty and impact. The literature on machine learning-based decoders for error-correcting codes is now very extensive, and this work does not meaningfully advance the state-of-the-art, either in methodology or insight.

-	Overall, the paper is solid but rather incremental. It would be more appropriate for a specialized or mid-tier venue rather than a top-tier conference like ICLR, which typically expects stronger conceptual contributions or more significant performance improvements (the proposed decoder does not beat state-of-the-art decoders).

[AC: The authors dispute that this reviewer may be biased by his prior review experience with the authors’ NeurIPS submission and his score should be discounted. No responses are provided to address his comments.]

Reviewer v6DD (4: marginally below the acceptance threshold; 3: You are fairly confident in your assessment.)

-	Inconsistencies, Typos, and Presentation Issues: The paper suffers from many typographical errors and presentation issues that reduce clarity and undermine its professional quality.

-	Figure-Text Inconsistency. In Figure 1(b), the Mamba block diagram is misleading. It correctly shows a SiLU activation for the  path, but omits the SiLU activation for the  path after the Conv block. This contradicts the text, as Eq (7) explicitly states .

-	Poor Presentation In Figure 1(b), the final output label () is poorly rendered and overlaps with the output arrow, demonstrating a lack of
attention to detail and reducing the diagram's readability.

-	Definition discrepancy in Mamba mask : The mathematical formulation in Eq (13) and (14) is misleading. The paper’s Mamba mask  is defined with dimensions , yet the paper applies it using indices  and  (model/state dimensions), as seen in  and . This indexing is dimensionally inconsistent with the mask's definition. 3. Lack of empirical runtime metrics: The conclusion states that ECCM achieves “low and improving inference speed”. However, the main text provides only the big-O complexity (Section 7.2) without any empirical latency, throughput, or FLOPs.

[AC: The reviewer raised serious concerns about the writing quality. The same concern is shared by another reviewer. The authors indicate that they have fixed all these issues.]

-	Runtime evidence: Were empirical measurements (e.g., samples/sec, latency, GPU memory) taken to support the “low and improving inference speed” claim? If so, please include them or provide a runtime benchmark appendix.

[AC: Some addition results are presented. However, no clear explanation about how the runtime measurements align with their theoretical complexity.]

-	Clarification on the SCL Baseline for Polar Code Comparison: In Appendix 12.2 and Table 4, the paper compares ECCM and CrossMPT against a classical "SCL (L=32)" baseline. Based on this, the authors conclude that ECCM "meaningfully closes the distance to the strong SCL baseline". However, the baseline is not specified as being CRC-Aided (CA-SCL), which is the standard high-performance SCL decoder configuration in modern literature. Could the authors clarify whether the "SCL (L=32)" baseline used in Table 4 includes a CRC? If it does not, how can the claim of closing the gap against a "strong SCL baseline" be justified, given that a ‘plain’ SCL is a significantly weaker baseline than the standard CA-SCL?

Reviewer j8n8 (4: marginally below the acceptance threshold; 4: You are confident in your assessment)

-	Despite the interesting idea, the paper has some clarity issues that make the method difficult to understand and implement correctly. These issues are detailed further in the questions section.

[AC: The same concern as the other reviewers.]

-	In Sec: 4.2 and 4.3 the authors state that if s_i=s the sample stops, and later outputs are “not calculated” and “not summed” in the loss. Could you add a short pseudocode block showing how this is implemented per sample within a batch during training and testing? And, could you please add a brief layer-usage summary showing where samples stop (i_"last" ) in particular, how many samples terminate before the final block?

[AC: This is a clarification comment. The authors have addressed this issue to the extent possible.]

-	For Table 2, could you explicitly state the number of blocks used in the ‘Transformer only’ and ‘Mamba only’ settings?

-	The authors train the Hybrid model with Adam at 2.5×10^-4and cosine annealing down to 10^-10(1000 batches/epoch). Could you briefly justify diverging from ECCT/CrossMPT LR/schedules? For Table 2, could you confirm that the ‘Transformer-only’ used the same LR/schedule? For comparability, please add a control run using the ECCT/CrossMPT learning rate and schedule.

[AC: These are clarification comments. The authors have addressed these issues to the extent possible.]

Reviewer RSom (6: marginally above the acceptance threshold. But would not mind if paper is rejected;  2: You are willing to defend your assessment,.)

-	The given architecture seems much more complex than prior art, and given that channel noise decoding and wireless systems are real-time systems, I would ask what the latency is and whether this high-performance algorithm is suitable for the applications the authors have envisioned.

[AC: The authors do not seem to have addressed this issue. ]

-	I also believe the paper would benefit from a hyperparameter sweep of the network, as it is the experiments seem to have been conducted for a given parameter set but we do not know if that is the optimal one. A balance of latency and quality would be necessary for these applications, and a Pareto plot of architecture/parameter count normalized by latency plotted against quality may be illustrative.

[AC: The authors argue that due to limited resources, they could not produce enough meaning results. However, some additional results are provided to address this issue.]

-	Would the authors say that this method remains unproven for real applications, given the complexity of transformers? Where do they see these approaches being used for real world deployments? Server-side decoder deployments may be unrealistic given latency and transmission requirements for wireless systems, is this intended for the edge?

[AC: The authors admit that this work is more of a foundational work and is currently unproven for commercial applications until specialized hardware is developed. Whether the authors’ responses are agreeable to the reviewer seems questionable.]

**Reviewer Concerns:**

See the summary section.

**Reviewer Scores:**

[AC: The initial reviews are mostly negative: 2, 4, 4, 6. The authors dispute that the reviewer giving the lowest score 2 may be biased by his prior experience with the authors’ previous submission to NeurIPS. The author with the highest score has low confidence. It also appears that his comments are not fully addressed. Given that some concerns remain not fully cleared and that none of the reviewers responded during the rebuttal period, this paper needs more work.]

---

### Decision · Program_Chairs · 2026-01-26

Reject